# Calcium-Bound S100P Protein Is a Promiscuous Binding Partner of the Four-Helical Cytokines

**DOI:** 10.3390/ijms231912000

**Published:** 2022-10-09

**Authors:** Alexey S. Kazakov, Evgenia I. Deryusheva, Maria E. Permyakova, Andrey S. Sokolov, Victoria A. Rastrygina, Vladimir N. Uversky, Eugene A. Permyakov, Sergei E. Permyakov

**Affiliations:** 1Institute for Biological Instrumentation, Pushchino Scientific Center for Biological Research of the Russian Academy of Sciences, 142290 Pushchino, Russia; 2Department of Molecular Medicine and USF Health Byrd Alzheimer’s Research Institute, Morsani College of Medicine, University of South Florida, Tampa, FL 33612, USA

**Keywords:** cytokine, S100 protein, S100P, protein–protein interaction

## Abstract

S100 proteins are multifunctional calcium-binding proteins of vertebrates that act intracellularly, extracellularly, or both, and are engaged in the progression of many socially significant diseases. Their extracellular action is typically mediated by the recognition of specific receptor proteins. Recent studies indicate the ability of some S100 proteins to affect cytokine signaling through direct interaction with cytokines. S100P was shown to be the S100 protein most actively involved in interactions with some four-helical cytokines. To assess the selectivity of the S100P protein binding to four-helical cytokines, we have probed the interaction of Ca^2+^-bound recombinant human S100P with a panel of 32 four-helical human cytokines covering all structural families of this fold, using surface plasmon resonance spectroscopy. A total of 22 cytokines from all families of four-helical cytokines are S100P binders with the equilibrium dissociation constants, *K_d_*, ranging from 1 nM to 3 µM (below the *K_d_* value for the S100P complex with the V domain of its conventional receptor, receptor for advanced glycation end products, RAGE). Molecular docking and mutagenesis studies revealed the presence in the S100P molecule of a cytokine-binding site, which overlaps with the RAGE-binding site. Since S100 binding to four-helical cytokines inhibits their signaling in some cases, the revealed ability of the S100P protein to interact with ca. 71% of the four-helical cytokines indicates that S100P may serve as a poorly selective inhibitor of their action.

## 1. Introduction

S100 proteins (reviewed in ref. [1,2,3]) are the most representative family of the Ca^2+^-binding proteins of the EF-hand superfamily, defined by the presence of a Ca^2+^-binding motif called the ‘EF-hand’ [4]: a 12-residue Ca^2+^-binding loop flanked by two α-helices (PROSITE [5] entry PDOC00018). Evolutionarily, the first S100 proteins (including S100P) emerged nearly 500 million years ago and are encountered exclusively in vertebrates [6]. The human S100 protein family includes 21 members (79–114 residues; excluding S100 fused-type proteins), comprised of an atypical low-affinity *N*-terminal EF-hand and a classical high-affinity C-terminal EF-hand motif connected via a flexible ‘hinge’ region [7]. With the exception of monomeric S100G, S100 proteins are considered (homo/hetero)dimeric, and some of them may form higher order oligomers [2,8]. Some S100 proteins possess Zn^2+^/Cu^2+^/Mn^2+^-binding sites [7,9] and undergo post-translational modifications [2,3]. The tissue/cell-specific expression of S100 proteins, their ability to localize in cytosol, nucleus, and extracellular space, and the capability to interact with a wide spectrum of targets (proteins, including receptor/membrane proteins, lipids, and nucleic acids) also contribute to their multifunctionality [1,2,10]. Despite their structural similarity, each of S100 proteins is functionally unique. Some of them are associated with cancer, inflammatory, autoimmune, cardiovascular, pulmonary, and neurodegenerative diseases, which makes them attractive diagnostic and therapeutic targets [11,12,13,14,15,16,17]. Upon their release into extracellular space, some S100 proteins act similarly to cytokines in an autocrine/paracrine manner via recognition of cell surface receptors, including RAGE, TLR4, ErbB1, ErbB3, ErbB4, CD36, CD68, CD147, CD166, neuroplastin-β, 5-HT_1B_, IL-10R, and SIRL-1 [2,10,18]. Furthermore, some of the extracellular S100 proteins are able to influence cytokine signaling via their direct binding; e.g., S100A1/A4/A6/B/P bind to IFN-β [10,19,20], distinct subsets of S100A1/A6/B/P interact with IL-6 family cytokines IL-11, OSM, CNTF, CT-1, and CLCF1 [21], S100A2/A6/P bind EPO [22], S100A4 binds to ErbB1 ligands [23], S100A13 interacts with IL1α/FGF1 [24,25], and S100B binds FGF2 [26]. Some of the S100-cytokine interactions could favor non-canonical secretion of both the interaction partners, as was shown for S100A13–IL1α/FGF1 [24,25].

It should be noted that most of the S100-cytokine interactions correspond to the four-helical cytokines, IFN-β, representatives of IL-6 family cytokines, and EPO. Among the cytokine-specific proteins, S100P recognizes the maximal number of the cytokines (see Table 1), indicating its poor selectivity towards the four-helical cytokines. To elucidate the selectivity of the S100P protein binding to four-helical cytokines, in the present work, we probed interaction of Ca^2+^-loaded S100P with a panel of 32 four-helical cytokines covering all their structural families, including “Short-chain cytokines” (SCOP [27] ID 4000852), “Long-chain cytokines” (SCOP ID 4000851), and “Interferons/interleukin-10 (IL-10)” (SCOP ID 4000854) (Table 2).

## 2. Results and Discussion

### 2.1. S100P Interaction with Specific Four-Helical Cytokines

To probe the selectivity of S100P binding to four-helical cytokines, 32 cytokines covering all families of this fold (Table 2) were immobilized on the surface of the SPR sensor chip by amine coupling, and 62 nM–16 µM solutions of Ca^2+^-loaded (1 mM CaCl_2_) recombinant human S100P were passed over the chip at 25 °C. The calcium concentration was chosen to be close to the level of free calcium in the serum of 1.1 mM. The temperature of 25 °C was used to ensure consistency with the previous SPR data on S100P interaction with other four-helical cytokines (Table 1). Ten cytokines indicated in Appendix A did not reveal specificity to S100P (data not shown). Meanwhile, the SPR sensograms for 22 cytokines exhibited the S100P concentration-dependent effects (Figure 1, Figure 2 and Figure 3).

The dissociation phases of the sensograms are biphasic revealing the existence of a relatively fast process and a much slower process. Both the processes are adequately described within the heterogeneous ligand model (1) (Figure 1, Figure 2 and Figure 3, Table 3), which was previously successfully used for description of the S100-cytokine interactions [10,19,20,21,22,28,29]. The SPR data for IL-13 are described by the one-site binding scheme. The lowest equilibrium dissociation constants, *K_d_*, range from 1 nM (THPO) to 3 µM (IL-10) (Table 3), which is below the *K_d_* value of 6 µM found earlier for the complex of Ca^2+^-bound S100P with the V domain of its receptor, RAGE [30].

Examination of the free energy changes accompanying the S100P–cytokine interactions (Figure 4) shows that the average affinities of S100P to the cytokines belonging to the different families of four-helical cytokines decrease in the following order: short-chain cytokines > long-chain cytokines > interferons/IL-10.

Examination of the IntAct [31] and BioGRID [32] databases shows the absence of the known soluble non-receptor extracellular proteins interacting with the following S100P-specific cytokines: IL-3, IL-5, IL-9, IL-13, IL-21, THPO, and IL-22. Most of them belong to the family of short-chain four-helical cytokines with the highest affinity to S100P (Figure 4A).

It should be noted that the S100P-specific cytokines are evolutionarily distant from each other. The pairwise sequence identities within each of their SCOP families, calculated using Clustal Omega 2.1, as implemented in EMBL-EBI service [33], lie in the range from 8% to 32% (the heterodimeric cytokines were excluded). The only exception is the GH–GH-V pair with the pairwise sequence identity of 93%. Therefore, the revealed S100P–cytokine interactions are mostly non-redundant.

The S100P-specific cytokines (Table 3) coupled to the SPR chip surface were readily regenerated after S100P binding upon calcium removal by passage of 20 mM EDTA solution pH 8.0 (data not shown). This fact points out the importance of the Ca^2+^-induced structural rearrangement for the efficient S100P binding to the cytokines. Since Ca^2+^ binding mostly induces solvent exposure of S100P residues of the ‘hinge’ between helices α2 and α3 [29,34], this region is likely to be involved in the cytokine recognition.

Considering that the equilibrium dimer dissociation constant for Ca^2+^-loaded S100P is about (64 ± 24) nM [35], the SPR estimates of its affinities to the cytokines, measured at S100P concentrations from 62 nM to 16 µM (Table 3), mostly correspond to the dimeric state of S100P. Meanwhile, the S100P dimer dissociation constant greatly exceeds basal serum S100P level of 1 nM [36], which suggests that the serum S100P is monomeric. We have shown previously that affinities of the four-helical cytokines IL-11 and IFN-β to the monomeric S100P exceed those to its dimeric state by 1.4–2.2 orders of magnitude [10,28,37]. Hence, S100P monomerization is likely to enhance its interaction with the other four-helical cytokines. In this case, *K_d_* values for some of the S100P–cytokine interactions (IL-3, IL-5, IL-9, THPO; IL-31, GH, PRL, and IL-26) may reach subnanomolar level or even lower (Figure 4), which is sufficient for the efficient cytokine binding to S100P at its basal serum level of 1 nM [36]. Furthermore, the blood S100P concentrations under pathological conditions may reach 5 nM [36]. Finally, the local concentrations of extracellular S100P in damaged S100P-producing tissues are expected to be even higher, thereby further promoting interactions of this protein with the four-helical cytokines.

The average blood concentrations of the most S100P-specific cytokines are normally below 0.1 nM (Appendix A), which is significantly lower than the basal blood S100P level of 1 nM [36]. Hence, these cytokines are quantitatively unable to affect signaling of the extracellular S100P via its receptor(s). The exceptions are GH, GH-V, PRL, and LEP, the average serum concentrations of which can approach the level of 1 nM or even exceed it (Appendix A). For instance, PRL concentration in the serum of pregnant women is in the range of 3.5–17.5 nM [38]. In these cases, cytokine binding to S100P is potentially able to alter the signaling of the latter. The same situation can occur under the pathological conditions accompanied by an increase in the concentrations of the S100P-specific cytokines to a (sub)nanomolar level, as observed for IL-9, IL-31, G-CSF, GH, GH-V, PRL, LEP, IL-24, and IL-26 (Appendix A).

S100P binding could modify signaling of the S100P-specific cytokines, as exemplified by inhibition of the IFN-β-induced suppression of the viability of MCF-7 breast cancer cells by S100A1/A4/B/P proteins [10,19,20]. Another opportunity is the facilitation of secretion of the S100P-specific cytokines due to the S100P binding, as previously shown for the four-helical cytokine CLCF1, which needs association with the CRLF1 for efficient secretion [39]. Similarly, S100A13 binding to cytokines IL-1α and FGF1 favors their non-canonical secretion [24,25].

The promiscuous binding of the wide spectrum of four-stranded cytokines by the S100P protein under the conditions of its quantitative excess indicates the possibility that S100P serves as a buffer for the cytokines, capable of absorbing their excess when the cytokine levels are excessively increased and releasing the cytokines under conditions of their depletion.

### 2.2. Modeling of the S100P–Cytokine Complexes

To elucidate molecular determinants of the revealed interactions between Ca^2+^-bound S100P and the wide spectrum of the four-helical cytokines (Table 3), models of the quaternary structures of their complexes have been built using ClusPro docking server [40] (except for the complexes with heterodimeric IL-27/IL-35, which lack fully resolved structures). The structure of Ca^2+^-loaded S100P dimer was extracted from its complex with V domain of RAGE (PDB entry 2MJW, Figure 5A), whereas the structures of the cytokines were taken from PDB (including complexes with their binding partners) or predicted using AlphaFold2 [41] (Appendix A, Figure 6). 

The predicted residues of the binding sites (included in five or more docking models) are shown in Appendix A and Figure 5A and Figure 6. The S100P residues predicted to be most commonly involved in the recognition of the four-helical cytokines studied include (19 cytokines out of 20): E5 of helix α1, and residues Y88 and F89 of helix α4. These residues were previously shown to bind V domain of RAGE (PDB entry 2MJW [30]), and predicted to interact with the short-chain cytokine EPO [22]. Furthermore, the residues Y88 and F89 were predicted to interact with IFN-β [19]. Analysis of the distributions of the predicted contact residues of S100P dimer over its amino acid sequence within models of S100P complexes with the members of specific families of the four-helical cytokines (Figure 5B) reveals that these families demonstrate very similar contact surfaces that include helices α1 and α4, and the ‘hinge’ region between helices α2 and α3. These S100P regions are united into the well-defined binding site, illustrated in Figure 5A. Notably, helices α1 and α4, and a ‘hinge’ of S100 proteins are frequently involved in the target recognition [42]. Namely, these elements of S100P’s secondary structure are engaged in the interaction with the V domain of RAGE (PDB entry 2MJW [30]). Moreover, the ‘hinge’ and helix α4 were earlier predicted to bind IFN-β [19]. Overall, the structural modelling points out the existence of considerable overlapping of the cytokine-binding site with that established for the RAGE domain.

To validate the modelling results, we have studied by SPR spectroscopy affinity of S100P mutants F89A and Δ42–47 (that lacks PGFLQS sequence in the flexible ‘hinge’) to 11 S100P-specific four-helical cytokines covering all families of this fold (Table 4). Despite the absence of the dramatic structural consequences of these mutations at the level of stabilities of their secondary, tertiary and quaternary structures (see Appendix A), with a few exceptions, both the mutants possess notably suppressed affinities to the cytokines (Table 4). The effect is more prominent for Δ42–47 mutant, lacking detectable interaction with nearly all cytokines studied, which implies that the respective *K_d_* values exceed 10^−4^ M. Therefore, the mutagenesis data support involvement of the ‘hinge’ and residue F89 in the recognition of most of the cytokines studied.

The long-chain cytokines are predicted to interact with Ca^2+^-bound S100P dimer via the residues of *N*-terminus and helices α1 and α3 (Figure 6B). The cytokines of interferons/IL-10 family, except for IL-26 with minimal number of the contact residues, are predicted to bind S100P dimer by helices α2 and α3, and some residues of C-terminus (Figure 6C). The predicted contact residues of the short-chain cytokines do not reveal evident regularity in their location (Figure 6A). Meanwhile, the contact residues of the γ_c_ family of cytokines (IL-9, IL-15, and IL-21 [43]) are located in the helices α1, α3, and α4, whereas the predicted S100P-binding site of THPO is similar to those of the long-chain cytokines PRL and LEP. Overall, the location of the predicted S100P-binding surfaces in the four-helical cytokines is variable and cytokine-dependent. 

Analysis of the cytokine tertiary structures in the complexes of some of the S100P-specific cytokines with their respective receptors shows partial overlapping of the receptor-binding sites with the predicted S100P-binding sites for GM-CSF, IL-3, IL-5, IL-15, IL-21, G-CSF, GH, IL-20, and IFN-ω1 (Appendix A). Among them, IL-3, IL-5, and GH belong to the group of cytokines with the highest affinity for S100P (Figure 4). Hence, S100P binding should interfere with the formation of the cytokine-receptor complexes. This conclusion is in line with the previous data on the inhibition of IFN-β signaling in MCF-7 cells by S100A1/A4/B/P [10,19,20].

It should be emphasized that the accuracy of the predictions of tertiary structures of the S100P–cytokine complexes based upon molecular docking is limited, since they do not take into consideration structural flexibility of the interaction partners. This is especially true in the case of the presence of the regions prone for disorder, including S100P (PDB entry 1J55 contains unresolved fragment 46–51), IL-15 (2Z3Q: unresolved fragment 76–80), THPO (1V7M, 1V7N: ~163 from 332 residues are ordered), G-CSF (DisProt [44] entry DP03184), LEP (1AX8: unresolved fragment 25–38), and IL-10 (2ILK: unresolved fragment 1–5). This is further illustrated by Figure 7 showing that although the majority of four-helical cytokines analyzed in this study are expected to be quite ordered, nevertheless, all of them are characterized by the noticeable levels of predicted intrinsic disorder. Therefore, the rather ordered nature of most of the cytokines validates the main conclusions of their modelling studies. On the other hand, the structural flexibility abundantly present in these proteins is likely to play an important role in their functionality. 

## 3. Materials and Methods

### 3.1. Materials

Human S100P protein and its F89A and Δ42–47 (lacks PGFLQS sequence) mutants were prepared in *E. coli* as previously described [28]. The cytokine samples used in the present work are listed in Table 2. Protein concentrations were measured spectrophotometrically using extinction coefficients at 280 nm calculated according to ref. [45].

Sodium acetate and ethanolamine were bought from Bio-Rad Laboratories, Inc. (Hercules, CA, USA) HEPES, sodium chloride, and SDS were from PanReac AppliChem (Barcelona, Spain). Potassium hydroxide, CaCl_2_, EDTA, and TWEEN 20 were purchased from Sigma–Aldrich Co., (Burlington, MA, USA). Tricine was from Helicon (Moscow, Russia). Glutaraldehyde was from Amersham Biosciences (Little Chalfont, UK). Coomassie brilliant blue R-250 was bought from Merck (Darmstadt, Germany).

### 3.2. Surface Plasmon Resonance Studies

Surface plasmon resonance (SPR) measurements of S100P affinity to cytokines at 25 °C were performed using ProteOn™ XPR36 protein interaction array system (Bio-Rad Laboratories, Inc., Hercules, CA, USA). Ligand (0.03 mg/ml cytokine) in 10 mM sodium acetate pH 4.5 buffer was immobilized on ProteOn™ GLH sensor chip surface (up to 9.000–14,000 RUs) by amine coupling, according to the manufacturer’s instructions. The remaining activated amine groups on the chip surface were blocked by 1 M ethanolamine solution. Analyte (62 nM–16 µM S100P) in a running buffer (10 mM HEPES, 150 mM NaCl, 1 mM CaCl_2_, 0.05% TWEEN 20, pH 7.4) was passed over the chip surface at a rate of 30 μl/min for 300 s, followed by flushing the chip with the running buffer for 2.400 s. Each double-referenced SPR sensogram was fitted according to either a one-site binding model (S100P–IL-13 interaction) or a *heterogeneous ligand* model (1), which assumes existence of two populations of the ligand (L_1_ and L_2_) that bind a single analyte molecule (A):

*k_a1_*

*k_a2_*

L_1_ + A→←L_1_AL_2_ + A→←L_2_A(1)
*k_d1_**K_d1_*

*k_d2_**K_d2_*


where *k_a_* and *k_d_* refer to kinetic association and dissociation constants, respectively, while *K_d_* are equilibrium dissociation constants (*k_d_*/*k_a_*). *K_d_* and *k_d_* values were evaluated for each analyte concentration using ProteOn Manager™ v.3.1 software (Bio-Rad Laboratories, Inc.), followed by averaging of the resulting values (*n* = 3–5). The standard deviations are indicated. The ligand was regenerated by passage of 20 mM EDTA solution pH 8.0 for 100 s. The free energy changes, accompanying the interaction between S100P and the cytokines were calculated as follows: ΔG_i_ = −RT ln(55.3/*K_di_*), i = 1,2.

### 3.3. Structural Classification of Cytokines

The cytokines were structurally classified according to the SCOP 2 database, build 1.0.6, updated on 29 June 2022 (https://scop2.mrc-lmb.cam.ac.uk, accessed on 1 July 2022 [27]). The cytokines studied belong to the “All alpha proteins” structural class, “4-helical cytokines” fold (SCOP ID 2001054; core: 4 helices, bundle, closed, left-handed twist, 2 crossover connections), “4-helical cytokines” superfamily (SCOP ID 3001717). The superfamily contains the following families: “Long-chain cytokines” (SCOP ID 4000851), “Short-chain cytokines” (SCOP ID 4000852) and “Interferons/interleukin-10 (IL-10)” (SCOP ID 4000854).

### 3.4. Modeling of the S100P–Cytokine Complexes

The models of tertiary structures of S100P–cytokine complexes were built using ClusPro docking server [40] mainly as previously described [22]. The structure of Ca^2+^-loaded human S100P dimer was extracted from the structure of its complex with V domain of RAGE (chains B, D of PDB [46] entry 2MJW). The tertiary structures of human cytokines have been taken from PDB or predicted using AlphaFold2 (https://alphafold.ebi.ac.uk/, accessed on 1 August 2022 [41]) (Appendix A). The signal peptides were excluded from the protein amino acid sequences for the predictions. In the case of the presence of unresolved regions in PDB structures of the proteins, their full-length structures were taken from tertiary structures of the proteins in complex with their binding partners. Distributions of the contact residues in the docking models over the protein sequences were calculated as described in ref. [22]. The contact residues included in 5 or more docking models were considered as the most probable residues of the binding site (numbering is according to the PDB entries). For distributions of the contact residues of Ca^2+^-loaded S100P dimer over its amino acid sequence within models of S100P complexes with representatives of specific families of four-helical cytokines 10 docking models were taken into account for each S100P–cytokine pair. The tertiary structures were drawn with molecular graphics system PyMOL v.2.5.0 (https://pymol.org/2/, accessed on 1 August 2022).

### 3.5. Comparison of Structural Properties of WT S100P and Its Mutants

Calcium removal from S100P samples was performed according to ref. [47]. Buffer conditions: 10 mM tricine-KOH, 1 mM EDTA-KOH, pH 7.4.

CD spectra of S100P samples (10 µM) were measured at 20 °C using JASCO J-810 spectropolarimeter (JASCO Inc., Tokyo, Japan), equipped with a Peltier-controlled cell holder (quartz cell with optical path length of 1 mm). The instrument was calibrated according to the manufacturer’s instruction. Bandwidth was 2 nm, averaging time 2 s, and accumulation 3. The spectral contribution of the buffer was subtracted from the protein spectra. 

Fluorescence spectra of S100P samples (10 µM) were measured using Cary Eclipse spectrometer (Varian Inc., Palo Alto, CA, USA), equipped with a Peltier-controlled cell holder (10 × 10 mm quartz cell). The sample temperature was monitored inside the cell. Fluorescence of S100P was excited at 280 nm; excitation and emission bandwidths were 5 nm; photomultiplier voltage of 800 V. Scanning of emission wavelengths from 290 nm to 380 nm, step 2 nm, averaging time 1 s. The emission spectra were corrected for spectral sensitivity of the spectrometer and described by a log-normal function [48] using LogNormal software (IBI RAS, Pushchino, Russia). The fluorescence emission maximum positions, λmax, were estimated from these fits. The temperature scans were performed stepwise upon heating, allowing the sample to equilibrate at each temperature for 3 min; average heating rate of 0.5 °C/min.

Crosslinking of decalcified S100P (67 µM) with 0.02% glutaraldehyde was performed at 20 °C for 16 h. The reaction was quenched by addition of SDS sample buffer. The resulting sample was analyzed by SDS–PAGE using 18% resolving gel, 5 µg of protein per lane [49]. The gels were stained with Coomassie brilliant blue R-250 and scanned using Molecular Imager PharosFX Plus System (Bio-Rad Laboratories, Inc., Hercules, CA, USA). The quantitation of each band was performed using Quantity One software.

### 3.6. Intrinsic Disorder Analysis of Human Four-Helical Cytokines

Intrinsic disorder propensities of human four-helical cytokines were evaluated by the PONDR^®^ VSL2 computational tool, which combines neural network predictors for short and long disordered regions [50] and which is one of the more accurate stand-alone disorder predictors [51,52,53]. To evaluate global disorder predisposition of a query protein, we calculated the percent of predicted disordered residues (i.e., residues with the disorder scores above 0.5) and also calculated the average disorder score as a protein length-normalized sum of all the per-residue disorder scores in a query protein. 

Important information on the global classification of disorder status of query proteins can be obtained using binary disorder predictors (i.e., computational tools that classify proteins as wholly ordered or wholly disordered), such as the charge-hydropathy (CH) plot [54,55] and the cumulative distribution function (CDF) plot [55,56]. CH-plot utilizes information on the absolute mean net charge and mean hydropathy to classify query proteins as proteins with substantial amounts of extended disorder (native coils and native pre-molten globules) or proteins with compact globular conformations (native molten globules and ordered proteins) [55,57]. CDF analysis uses the PONDR outputs to discriminate all types of disorder (native coils, native molten globules and native pre-molten globules) from ordered proteins [55]. As a result, the combined CH-CDF plot provides an opportunity for unique assessment of intrinsic disorder in several categories, providing a means for predictive classification of proteins into structurally different classes [56,58,59]. To generate a corresponding CH-CDF plot, the coordinates of a query protein are calculated as a distance from the boundary in the CH-plot (Y-coordinate) and an average distance of the respective CDF curve from the CDF boundary (X-coordinate). Proteins are then classified based on their positions within the quadrants of the CH-CDF plot. Here, the lower-right quadrant (Q1) includes ordered proteins (i.e., those predicted as ordered and compact by both CDF and CH); the lower-left quadrant (Q2) contains proteins predicted to be disordered by CDF but compact by CH-plot (i.e., native molten globules or hybrid proteins containing sizable levels of order and disorder); the upper-left quadrant (Q3) contains proteins predicted to be disordered by both methods (i.e., proteins with extended disorder, such as native coils and native pre-molten globules); and the upper-right quadrant (Q4) contains proteins predicted to be disordered by CH-plot but ordered by CDF [58].

## 4. Conclusions

Among the 41 four-helical cytokines studied to date with regard to their affinity to the S100P protein (Table 1, Table 2 and Appendix A), only 12 cytokines did not show specificity to S100P (Appendix A). Thus, ca. 71% of the four-helical cytokines exhibit affinity to Ca^2+^-bound S100P dimer exceeding that for the V domain of its receptor, RAGE (Table 1 and Table 3). Since the S100P-specific cytokines are evolutionarily distant from each other, except for the GH–GH-V pair, the S100P–cytokine interactions are mostly non-redundant. Considering that the S100P monomerization is expected to favor its interaction with the four-helical cytokines, the fraction of the S100P-specific cytokines is likely to be even higher. Therefore, S100P represents a unique case of the promiscuous binding partner for the most of four-helical cytokines. The molecular docking and the mutagenesis study point out some structural basics of this phenomenon. The cytokine-binding sites of the S100P protein seem to overlap with its RAGE-binding site, and include residues of helices α1 and α4, and the ‘hinge’ region (Figure 5). Meanwhile, regularities of the location of S100P-binding sites of the cytokines remain fairly enigmatic (Figure 6). Nevertheless, in some cases these sites seem to overlap with the known receptor-binding sites of the cytokines (Appendix A), thereby indicating that binding of the extracellular S100P should interfere with their signaling. Binding of some of the cytokines to S100P is potentially able to modulate its signaling, especially under the pathological conditions accompanied by an increase in concentrations of the cytokines. Binding of the intracellular S100P could promote non-canonical secretion of the S100P-specific cytokines, similarly to the several reported cases [24,25,39]. Finally, quantitative excess of the S100P protein suggests that it could serve as a buffer for the multiple four-helical cytokines, stabilizing their free concentrations.

S100P is considered as one of the most promiscuous members of the S100 protein family, able to bind numerous ligands with a high affinity, and sharing interaction partners with other S100 proteins [60,61]. Therefore, one may expect that other representatives of the S100 protein family are cross-reactive with the same four-helical cytokines, as previously shown for IL-6 family [21]. Further studies should verify this hypothesis, provide more molecular details of the S100-cytokine interactions, help us to achieve a better understanding of the structural basis of their selectivity, and establish their functional significance.

## Figures and Tables

**Figure 1 ijms-23-12000-f001:**
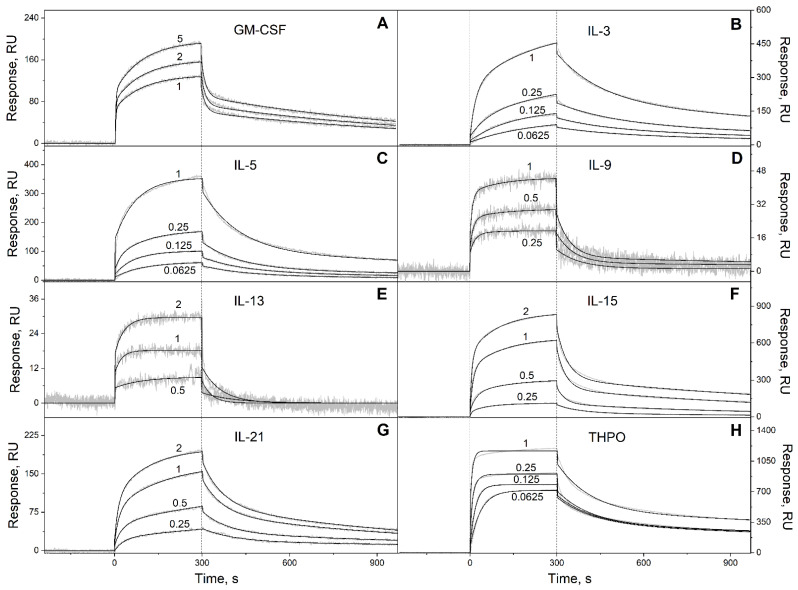
Kinetics of the interaction of Ca^2+^-loaded S100P with specific short-chain four-helical cytokines shown in Table 2 at 25 °C, monitored by SPR spectroscopy using S100P as an analyte and the cytokines as a ligand immobilized on the sensor chip surface by amine coupling. (**A**): GM-CSF; (**B**): IL-3; (**C**): IL-5; (**D**): IL-9; (**E**): IL-13; (**F**): IL-15; (**G**): IL-21; (**H**): THPO. Buffer conditions: 10 mM HEPES, 150 mM NaCl, 1 mM CaCl_2_, 0.05% TWEEN 20, pH 7.4. The vertical dotted lines mark the association phase, followed by the dissociation phase. Molar analyte concentrations (µM) are indicated for the sensograms. The gray curves are experimental, while the black curves are theoretical, calculated according to the *heterogeneous ligand* model (1) or one-site binding model, in the case of IL-13 (see Table 3).

**Figure 2 ijms-23-12000-f002:**
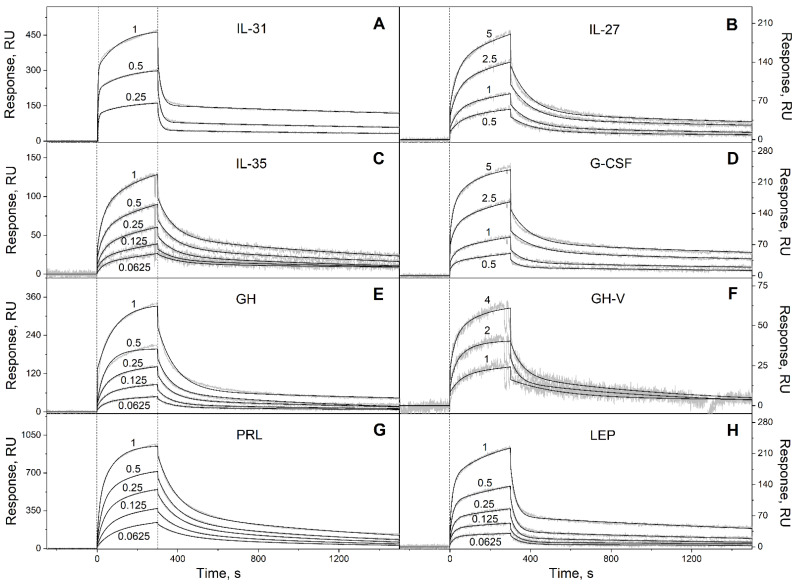
Kinetics of the interaction of Ca^2+^-loaded S100P with specific long-chain four-helical cytokines shown in Table 2 at 25 °C, monitored by SPR spectroscopy using S100P as an analyte and the cytokines as a ligand immobilized on the sensor chip surface by amine coupling. (**A**): IL-31; (**B**): IL-27; (**C**): IL-35; (**D**): G-CSF; (**E**): GH; (**F**): GH-V; (**G**): PRL; (**H**): LEP. Buffer conditions: 10 mM HEPES, 150 mM NaCl, 1 mM CaCl_2_, 0.05% TWEEN 20, pH 7.4. The vertical dotted lines mark the association phase, followed by the dissociation phase. Molar analyte concentrations (µM) are indicated for the sensograms. The gray curves are experimental, while the black curves are theoretical, calculated according to the *heterogeneous ligand* model (1) (see Table 3).

**Figure 3 ijms-23-12000-f003:**
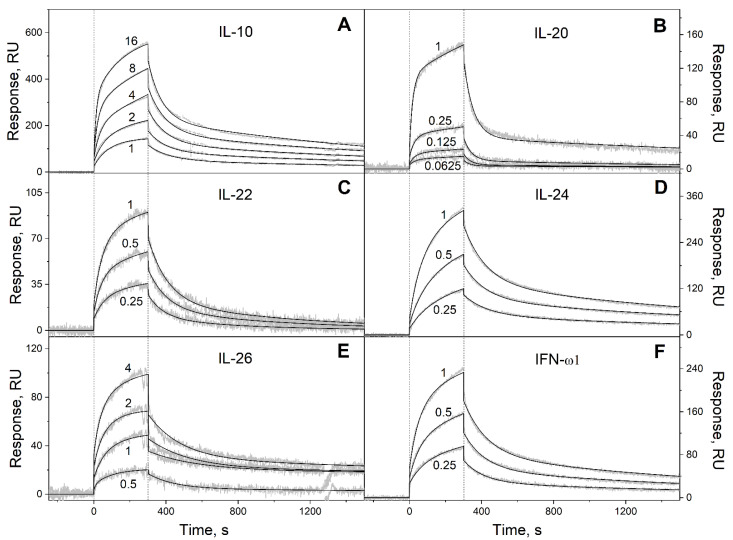
Kinetics of the interaction of Ca^2+^-loaded S100P with specific four-helical cytokines of Interferons/IL-10 family shown in Table 2 at 25 °C, monitored by SPR spectroscopy using S100P as an analyte and the cytokines as a ligand immobilized on the sensor chip surface by amine coupling. (**A**): IL-10; (**B**): IL-20; (**C**): IL-22; (**D**): IL-24; (**E**): IL-26; (**F**): IFN-ω1. Buffer conditions: 10 mM HEPES, 150 mM NaCl, 1 mM CaCl_2_, 0.05% TWEEN 20, pH 7.4. The vertical dotted lines mark the association phase, followed by the dissociation phase. Molar analyte concentrations (µM) are indicated for the sensograms. The gray curves are experimental, while the black curves are theoretical, calculated according to the *heterogeneous ligand* model (1) (see Table 3).

**Figure 4 ijms-23-12000-f004:**
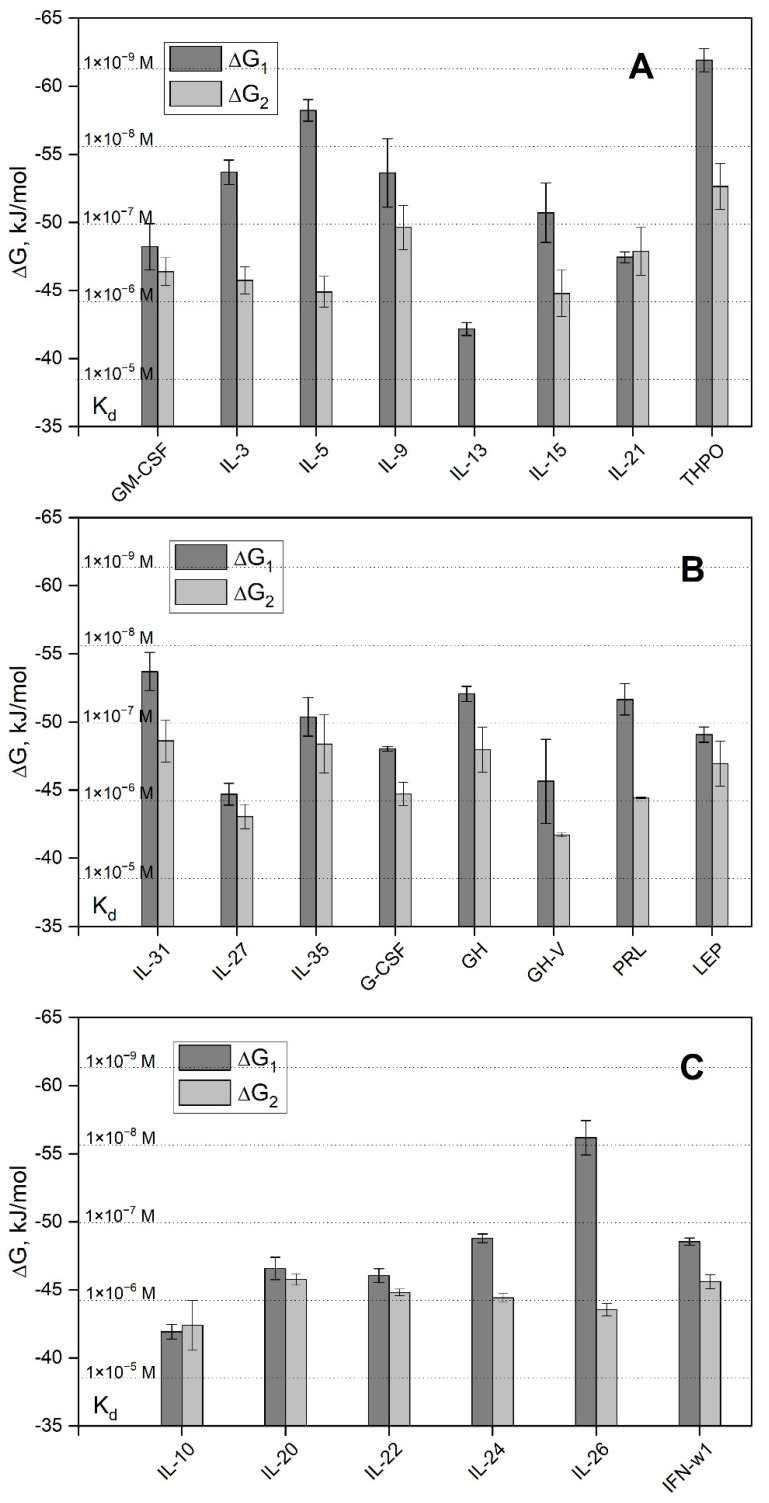
The free energy changes, accompanying the interaction between Ca^2+^-loaded S100P and the four-helical cytokines of short-chain (panel **A**), long-chain (**B**) or interferons/IL-10 (**C**) families at 25 °C, calculated from the SPR data shown in Table 3: ΔG_i_ = −RT ln(55.3/*K_di_*), i = 1,2. The scale of *K_d_* values is indicated.

**Figure 5 ijms-23-12000-f005:**
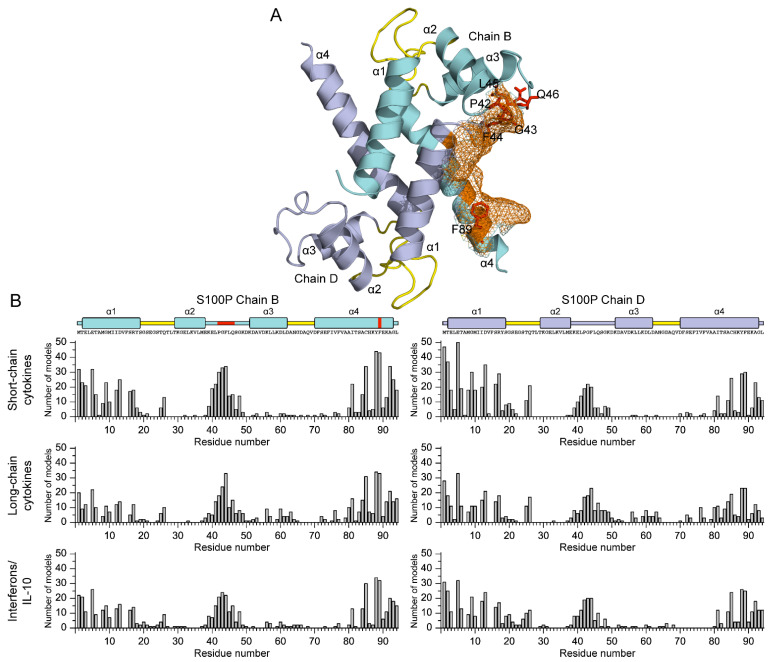
(**A**) Tertiary structure of Ca^2+^-loaded S100P dimer (PDB entry 2MJW): chains B and D are highlighted in cyan and gray, respectively; the α-helices are labelled as α1-α4; the Ca^2+^-binding loops are yellow-colored. The residues predicted to constitute the cytokine-binding site are shown as orange mesh surface: residues P42, G43, F44, C85, Y88, and F89 (chain B), and residues M1 and E5 (chain D). The residues P42, G43, F44, L45, Q46, and F89 (see Table 4) of chain B are depicted as red balls and sticks. (**B**) Distributions of the predicted contact residues of Ca^2+^-loaded S100P dimer over its amino acid sequence within models of S100P complexes with representatives of specific families of four-helical cytokines (see Appendix A). Ten docking models were taken into account for each S100P–cytokine pair. The boundaries of secondary structure elements of S100P were taken from PDB entry 2MJW; the residues P42, G43, F44, L45, Q46, and F89 (see Table 4) of chain B are indicated in red.

**Figure 6 ijms-23-12000-f006:**
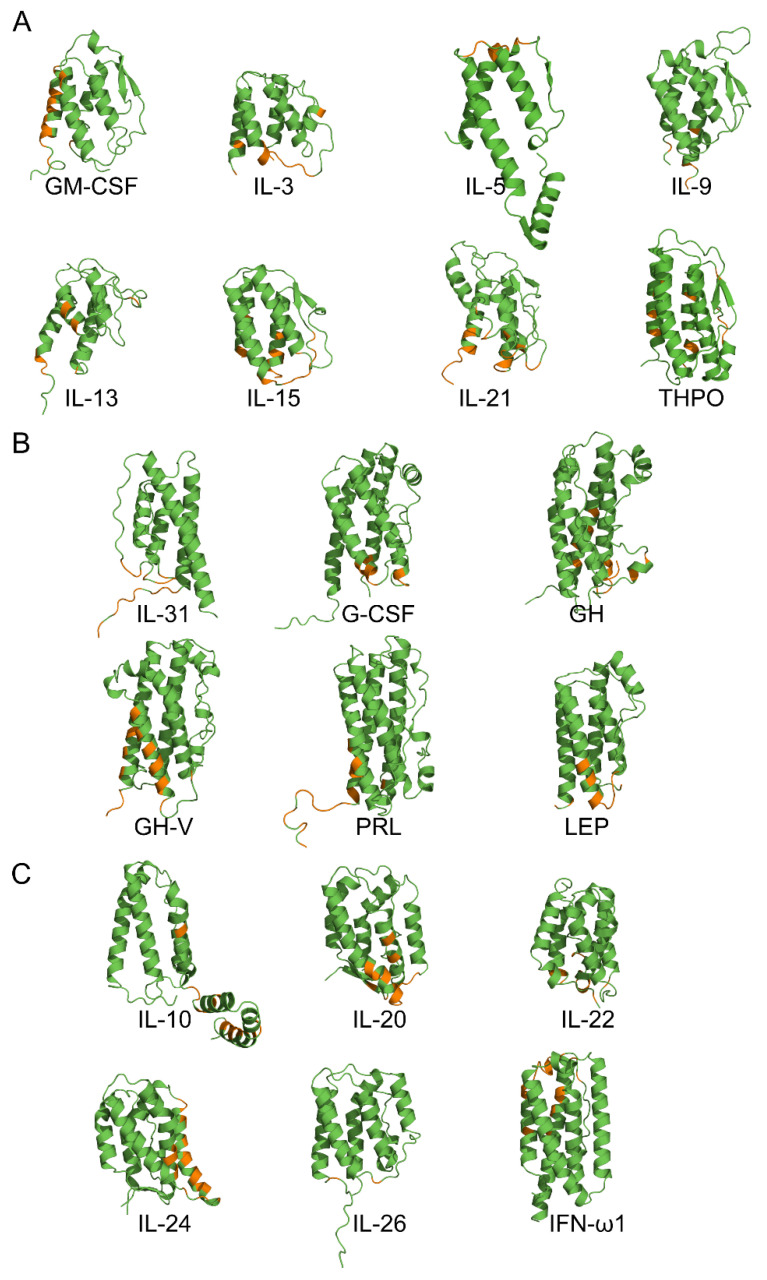
The tertiary structures of the four-helical cytokines of short-chain (panel **A**), long-chain (**B**) and interferons/IL-10 (**C**) families (Appendix A) used for the structural modeling of their complexes with Ca^2+^-loaded S100P dimer by ClusPro docking server [40] (only one subunit is shown for IL-5). The contact residues included in five or more docking models were considered as the most probable residues of the binding site (orange-colored). *N*-terminus is located in the lower-left corner for each cytokine.

**Figure 7 ijms-23-12000-f007:**
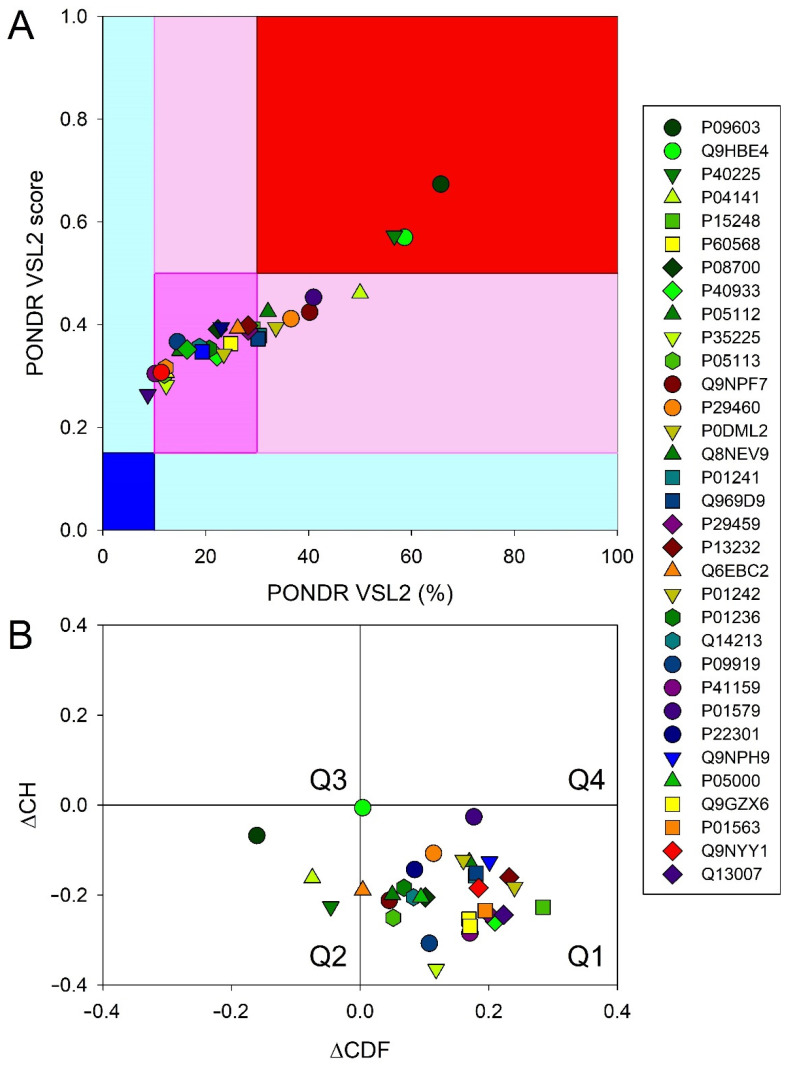
Global intrinsic disorder analysis of the human four-helical cytokines. (**A**) The PONDR^®^ VSL2 output, where the PONDR^®^ VSL2 score is the average disorder score of a query protein, and PONDR^®^ VSL2 (%) is a percentage of the predicted disordered residues with the disorder scores above 0.5. Color blocks indicate regions in which proteins are mostly ordered (blue and light blue), moderately disordered (pink and light pink), or mostly disordered (red). If the two parameters agree, the corresponding part of background is dark (blue, pink, or red), whereas light blue and light pink reflect areas in which only one of these criteria applies. (**B**) CH-CDF plot combining outputs of the charge-hydropathy (CH) and cumulative distribution function (CDF) analyses. The Y-coordinate is calculated as the distance of the corresponding protein from the boundary in the CH plot. The X-coordinate is calculated as the average distance of the corresponding protein’s CDF curve from the CDF boundary. The quadrant that the protein is located determines its disorder-based classification: Q1, protein predicted to be ordered by both tools (i.e., mostly ordered proteins); Q2, protein predicted to be ordered by CH-plot and disordered by CDF (i.e., native molten globules); Q3, protein predicted to be disordered by both tools (i.e., mostly disordered proteins); Q4, protein predicted to be disordered by CH-plot and ordered by CDF.

**Table 1 ijms-23-12000-t001:** The literature data on parameters of the heterogeneous ligand model (1), describing the SPR data on kinetics of interaction between Ca^2+^-loaded S100P and four-helical cytokines at 25 °C (cytokine immobilization on the sensor chip surface by amine coupling).

Cytokine	*K_d1_*, M	*K_d2_*, M	Reference
Short-chain cytokines			
EPO	(5.4 ± 1.2) × 10^−7^	(1.8 ± 0.5) × 10^−6^	[22]
Long-chain cytokines			
IL-11	(3.2 ± 0.3) × 10^−8^	(2.88 ± 0.01) × 10^−7^	[28]
Cardiotrophin-like cytokine factor 1	(8.1 ± 2.6) × 10^−8^	(1.4 ± 0.8) × 10^−7^	[21]
Ciliary neurotrophic factor	(1.0 ± 0.6) × 10^−7^	(1.1 ± 0.8) × 10^−7^	[21]
Cardiotrophin-1	(1.9 ± 0.5) × 10−^8^	(9.8 ± 2.7) × 10^−7^	[21]
Oncostatin-M	(7.0 ± 4.2) × 10^−7^	(2.0 ± 0.9) × 10^−6^	[21]
Interferons/IL-10			
IFN-β	(5.34 ± 0.10) × 10^−8^	(6.1 ± 2.3) × 10^−7^	[10]

**Table 2 ijms-23-12000-t002:** The four-helical cytokine samples studied in the present work with regard to their affinity to Ca^2+^-bound S100P.

Full Name	Abbreviation	UniProt ID	Manufacturer	Cat. Number	Source
Short-chain cytokines					
Macrophage colony-stimulating factor 1	M-CSF	P09603	PeproTech	300-25	*E. coli*
Granulocyte-macrophage colony-stimulating factor	GM-CSF	P04141	PeproTech	300-03	*E. coli*
Interleukin-2	IL-2	P60568	PeproTech	AF-200-02	*E. coli*
Interleukin-3	IL-3	P08700	SCI-Store (Russia)	PSG160-10	CHO
Interleukin-4	IL-4	P05112	PeproTech	AF-200-04	*E. coli*
Interleukin-5	IL-5	P05113	PeproTech	200-05	*E. coli*
Interleukin-9	IL-9	P15248	PeproTech	200-09	*E. coli*
Interleukin-13	IL-13	P35225	PeproTech	200-13	*E. coli*
Interleukin-15	IL-15	P40933	PeproTech	200-15	*E. coli*
Interleukin-21	IL-21	Q9HBE4	PeproTech	200-21	*E. coli*
Thrombopoietin	THPO	P40225	SCI-Store (Russia)	PSG090-10	CHO
Long-chain cytokines					
Interleukin-7	IL-7	P13232	SCI-Store (Russia)	PSG240-10	CHO
Interleukin-31	IL-31	Q6EBC2	PeproTech	200-31	*E. coli*
Granulocyte colony-stimulating factor	G-CSF	P09919	Pharmstandard (Russia)	n/a	*E. coli*
Somatotropin	GH	P01241	PeproTech	AF-100-40	*E. coli*
Growth hormone variant	GH-V	P01242	R&D Systems	7668-GH/CF	*E. coli*
Prolactin	PRL	P01236	PeproTech	100-07	*E. coli*
Leptin	LEP	P41159	PeproTech	AF-300-27	*E. coli*
Thymic stromal lymphopoietin	TSLP	Q969D9	PeproTech	300-62	*E. coli*
Chorionic somatomammotropin hormone 1	PL	P0DML2	R&D Systems	5757-PL/CF	CHO
Interleukin-12	IL-12	P29459 * and P29460	PeproTech	200-12H	HEK293
Interleukin-23	IL-23	Q9NPF7 * and P29460	PeproTech	200-23	Hi-5
Interleukin-27	IL-27	Q8NEV9 * and Q14213	PeproTech	200-38	HEK293
Interleukin-35	IL-35	P29459 * and Q14213	PeproTech	200-37	HEK293
Interferons/IL-10					
Interleukin-10	IL-10	P22301	PeproTech	AF-200-10	*E. coli*
Interleukin-20	IL-20	Q9NYY1	PeproTech	200-20	*E. coli*
Interleukin-22	IL-22	Q9GZX6	PeproTech	200-22	*E. coli*
Interleukin-24	IL-24	Q13007	PeproTech	200-35	CHO
Interleukin-26	IL-26	Q9NPH9	R&D Systems	1375-IL/CF	*E. coli*
Interferon α-2	IFN-α2	P01563	Vector-Medica (Russia)	n/a	*E. coli*
Interferon γ	IFN-γ	P01579	Pharmaclon (Russia)	n/a	*E. coli*
Interferon ω-1	IFN-ω1	P05000	PeproTech	300-02J	*E. coli*

* denotes the chain used for SCOP 2 [27] family assignment; n/a, not applicable.

**Table 3 ijms-23-12000-t003:** Parameters of the *heterogeneous ligand* model (1), describing the SPR data on kinetics of interaction between Ca^2+^-loaded S100P and specific four-helical cytokines shown in Table 2 at 25 °C (see Figure 1, Figure 2 and Figure 3).

Cytokine	*k_d1_*, s^−1^	*K_d1_*, M	*k_d2_*, s^−1^	*K_d2_*, M
Short-chain cytokines				
GM-CSF	(1.13 ± 0.23) × 10^−3^	(2.45 ± 1.45) × 10^−7^	(6.72 ± 2.13) × 10^−2^	(4.50 ± 1.75) × 10^−7^
IL-3	(6.27 ± 0.38) × 10^−4^	(2.32 ± 0.81) × 10^−8^	(5.29 ± 0.86) × 10^−3^	(5.79 ± 2.19) × 10^−7^
IL-5	(6.12 ± 1.12) × 10^−4^	(3.66 ± 1.13) × 10^−9^	(7.12 ± 0.77) × 10^−3^	(8.33 ± 3.58) × 10^−7^
IL-9	(5.31 ± 1.69) × 10^−4^	(3.47 ± 2.66) × 10^−8^	(1.70 ± 0.48) × 10^−2^	(1.36 ± 0.78) × 10^−7^
IL-13 *	(1.83 ± 1.09) × 10^−2^	(2.30 ± 0.44) × 10^−6^	n/a	n/a
IL-15	(1.09 ± 0.25) × 10^−3^	(1.02 ± 0.72) × 10^−7^	(2.72 ± 1.03) × 10^−2^	(9.80 ± 5.84) × 10^−7^
IL-21	(9.28 ± 2.85) × 10^−4^	(2.72 ± 0.43) × 10^−7^	(1.26 ± 0.26) × 10^−2^	(2.85 ± 1.73) × 10^−7^
THPO	(4.09 ± 0.49) × 10^−4^	(8.34 ± 2.79) × 10^−10^	(8.29 ± 0.38) × 10^−3^	(4.12 ± 2.44) × 10^−8^
Long-chain cytokines				
IL-31	(2.30 ± 0.52) × 10^−4^	(2.52 ± 1.30) × 10^−8^	(5.83 ± 0.79) × 10^−2^	(2.02 ± 1.12) × 10^−7^
IL-27 ^#^	(9.94 ± 0.75) × 10^−3^	(8.59 ± 2.67) × 10^−7^	(3.57 ± 0.64) × 10^−4^	(1.69 ± 0.59) × 10^−6^
IL-35 ^#^	(5.31 ± 1.31) × 10^−4^	(9.62 ± 4.92) × 10^−8^	(1.20 ± 0.13) × 10^−2^	(2.55 ± 1.77) × 10^−7^
G-CSF	(3.15 ± 0.46) × 10^−4^	(2.13 ± 0.15) × 10^−7^	(1.54 ± 0.72) × 10^−2^	(8.54 ± 2.80) × 10^−7^
GH	(6.69 ± 2.77) × 10^−4^	(4.29 ± 0.93) × 10^−8^	(1.42 ± 0.18) × 10^−2^	(2.66 ± 1.54) × 10^−7^
GH-V	(1.77 ± 1.05) × 10^−3^	(1.04 ± 0.88) × 10^−6^	(1.81 ± 0.71) × 10^−2^	(2.73 ± 0.15) × 10^−6^
PRL	(9.81 ± 0.84) × 10^−4^	(5.46 ± 2.35) × 10^−8^	(1.03 ± 0.10) × 10^−2^	(9.06 ± 0.17) × 10^−7^
LEP	(4.52 ± 0.77) × 10^−4^	(1.43 ± 0.32) × 10^−7^	(3.07 ± 0.75) × 10^−2^	(4.04 ± 2.35) × 10^−7^
Interferons/IL-10				
IL-10	(1.06 ± 0.17) × 10^−2^	(2.55 ± 0.54) × 10^−6^	(5.19 ± 0.43) × 10^−4^	(2.66 ± 1.68) × 10^−6^
IL-20	(4.42 ± 0.97) × 10^−4^	(4.03 ± 1.29) × 10^−7^	(2.46 ± 0.51) × 10^−2^	(5.37 ± 0.90) × 10^−7^
IL-22	(7.64 ± 5.76) × 10^−3^	(4.82 ± 0.96) × 10^−7^	(2.76 ± 1.93) × 10^−3^	(7.80 ± 0.77) × 10^−7^
IL-24	(4.83 ± 0.21) × 10^−4^	(1.58 ± 0.20) × 10^−7^	(7.94 ± 0.34) × 10^−3^	(9.18 ± 1.24) × 10^−7^
IL-26	(1.44 ± 0.78) × 10^−4^	(9.02 ± 4.24) × 10^−9^	(5.09 ± 1.38) × 10^−3^	(1.32 ± 0.24) × 10^−6^
IFN-ω1	(5.71 ± 0.09) × 10^−4^	(1.74 ± 0.19) × 10^−7^	(8.39 ± 0.56) × 10^−3^	(5.78 ± 1.18) × 10^−7^

***** one-site binding model is used; ^#^ heterodimeric cytokines; n/a, not applicable.

**Table 4 ijms-23-12000-t004:** Parameters of the heterogeneous ligand model (1), describing the SPR data on kinetics of interaction between Ca^2+^-loaded wild-type S100P or its mutants and the four-helical cytokines shown in Table 3 at 25 °C.

*Cytokine\S100P*	*Wild-Type*		*F89A*		*Δ42–47*
	*K_d1_, M*	*K_d2_, M*	*K_d1_, M*	*K_d1_, M*	*K_d_, M*
*Short-chain cytokines*					
GM-CSF	(2.45 ± 1.45) × 10^−7^	(4.50 ± 1.75) × 10^−7^	n.d.		n.d.
IL-3	(2.32 ± 0.81) × 10^−8^	(5.79 ± 2.19) × 10^−7^	n.d.		n.d.
IL-5	(3.66 ± 1.13) × 10^−9^	(8.33 ± 3.58) × 10^−7^	n.d.		n.d.
IL-9	(3.47 ± 2.66) × 10^−8^	(1.36 ± 0.78) × 10^−7^	n.d.		n.d.
IL-13 *	(2.30 ± 0.44) × 10^−6^	n/a	>10^−4^		>10^−4^
IL-15	(1.02 ± 0.72) × 10^−7^	(9.80 ± 5.84) × 10^−7^	(1.57 ± 0.57) × 10^−7^	(2.83 ± 0.34) × 10^−7^	>10^−4^
IL-21	(2.72 ± 0.43) × 10^−7^	(2.85 ± 1.73) × 10^−7^	(4.01 ± 1.64) × 10^−7^	(9.50 ± 4.52) × 10^−7^	>10^−4^
THPO	(8.34 ± 2.79) × 10^−10^	(4.12 ± 2.44) × 10^−8^	n.d.		n.d.
*Long-chain cytokines*					
IL-31	(2.52 ± 1.30) × 10^−8^	(2.02 ± 1.12) × 10^−7^	(2.02 ± 0.59) × 10^−7^	(1.18 ± 0.53) × 10^−6^	>10^−4^
IL-27 ^#^	(8.59 ± 2.67) × 10^−7^	(1.69 ± 0.59) × 10^−6^	n.d.		n.d.
IL-35 ^#^	(9.62 ± 4.92) × 10^−8^	(2.55 ± 1.77) × 10^−7^	>10^−4^		>10^−5^
G-CSF	(2.13 ± 0.15) × 10^−7^	(8.54 ± 2.80) × 10^−7^	n.d.		n.d.
GH	(4.29 ± 0.93) × 10^−8^	(2.66 ± 1.54) × 10^−7^	(2.92 ± 2.66) × 10^−5^	(1.38 ± 0.52) × 10^−4^	>10^−4^
GH-V	(1.04 ± 0.88) × 10^−6^	(2.73 ± 0.15) × 10^−6^	>10^−4^		>10^−4^
PRL	(5.46 ± 2.35) × 10^−8^	(9.06 ± 0.17) × 10^−7^	(6.89 ± 3.20) × 10^−7^	(3.12 ± 0.50) × 10^−6^	>10^−4^
LEP	(1.43 ± 0.32) × 10^−7^	(4.04 ± 2.35) × 10^−7^	(4.37 ± 1.71) × 10^−6^	(3.57 ± 1.77) × 10^−5^	>10^−4^
*Interferons/IL-10*					
IL-10	(2.55 ± 0.54) × 10^−6^	(2.66 ± 1.68) × 10^−6^	(4.74 ± 3.41) × 10^−6^	(6.05 ± 1.47) × 10^−6^	>10^−4^
IL-20	(4.03 ± 1.29) × 10^−7^	(5.37 ± 0.90) × 10^−7^	n.d.		n.d.
IL-22	(4.82 ± 0.96) × 10^−7^	(7.80 ± 0.77) × 10^−7^	n.d.		n.d.
IL-24	(1.58 ± 0.20) × 10^−7^	(9.18 ± 1.24) × 10^−7^	n.d.		n.d.
IL-26	(9.02 ± 4.24) × 10^−9^	(1.32 ± 0.24) × 10^−6^	>10^−4^		>10^−4^
IFN-ω1	(1.74 ± 0.19) × 10^−7^	(5.78 ± 1.18) × 10^−7^	n.d.		n.d.

***** one-site binding model is used; ^#^ heterodimeric cytokines; n/a, not applicable; n.d., not determined.

## Data Availability

The data supporting the reported results can be found at the Laboratory of New Methods in Biology of the Institute for Biological Instrumentation, Pushchino Scientific Center for Biological Research of the Russian Academy of Sciences, 142290 Pushchino, Russia.

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
