# Peer review of "Calcium-Bound S100P Protein Is a Promiscuous Binding Partner of the Four-Helical Cytokines"

_ijms, 2022, doi:10.3390/ijms231912000_

Round 1

Reviewer 1 Report

The manuscript "Calcium-bound S100P Protein is a Promiscuous Binding Partner of the Four-Helical Cytokines" by Kazakov et al., describes a surface plasmon resonance study of the interaction of calcium-bound recombinant S100P protein with a myriad of four-helical human cytokines, determining equilibrium dissociation constants. The work is completed with docking and mutagenesis studies of the S100P-cytokine binding site. The article is well written and the experimental procedures are clearly described. Although some of the results presented here are merely confirmatory and some of the conclusions are quite speculative, the work as a whole could be an interesting addition to the field, especially considering the validation of the binding site performed with mutants F89A and Δ42-47 of S100P. Some minor concerns are indicated below.

1. SPR experiments are performed at 25ºC. I guess that ideally this temperature should be the physiological 37ºC, but I understand that there might be technical restrictions. The authors should at least comment on this and how it affects their conclusions.

2. The authors state that models of the quaternary structures of Ca2+-bound S100P and four-helical cytokines have been built except for heterodimeric IL-27/IL-35 (page 9, line 191). They should clarify the reason for this exception.  

3. On page 13, lines 255-260, the authors speculate that S100P binding should interfere with the formation of cytokine-receptor complexes. This of course will depend on specific concentrations and Kd’s at physiological conditions. The authors should consider being more specific and, if possible, quantitative on this point.

4. I might have missed something, but, at least to me, it is not very clear to what extent the analysis of disordered regions depicted on Figure 7 adds to the conclusions of the paper. Authors should consider being more specific on the main conclusion of this figure, this possibly being that most of the cytokines analyzed seem to be quite ordered, thus validating the main conclusions of their modelling studies.

5. Minor English editing is needed, especially regarding the unnecessary use of some commas (“it should be noted, that…”, “it should be emphasized, that…”, page 2, line 61; page 7, line 132; page 13, line 261) and the definite article “the” (“the both processes”, “the both mutants”, “the most of the cytokines”, among many other examples). Ca2+ should be used instead of Ca2+ and ºC instead of ºC. There are also some repeated lines (page 2, line 82 to page 3, line 87) and redundancies (first paragraph on page 13).

Author Response

The manuscript "Calcium-bound S100P Protein is a Promiscuous Binding Partner of the Four-Helical Cytokines" by Kazakov et al., describes a surface plasmon resonance study of the interaction of calcium-bound recombinant S100P protein with a myriad of four-helical human cytokines, determining equilibrium dissociation constants. The work is completed with docking and mutagenesis studies of the S100P-cytokine binding site. The article is well written and the experimental procedures are clearly described. Although some of the results presented here are merely confirmatory and some of the conclusions are quite speculative, the work as a whole could be an interesting addition to the field, especially considering the validation of the binding site performed with mutants F89A and Δ42-47 of S100P. Some minor concerns are indicated below.

  1. SPR experiments are performed at 25ºC. I guess that ideally this temperature should be the physiological 37ºC, but I understand that there might be technical restrictions. The authors should at least comment on this and how it affects their conclusions.

Response: The temperature of 25°C was used to ensure consistency with the previous SPR data on S100P interaction with other four-helical cytokines (Table 1). We have added this clarification to the first paragraph of section 2.1. Given the high thermal stability of S100P (Cell Calcium, 2019, v.80, p.152-159) and the cytokines, the temperature difference of 12°C is unlikely to noticeably affect their interaction.

  1. The authors state that models of the quaternary structures of Ca2+-bound S100P and four-helical cytokines have been built except for heterodimeric IL-27/IL-35 (page 9, line 191). They should clarify the reason for this exception.

Response: We excluded IL-27 and IL-35 from the modelling, since they lack fully resolved structures. We have included this clarification into the text.

  1. On page 13, lines 255-260, the authors speculate that S100P binding should interfere with the formation of cytokine-receptor complexes. This of course will depend on specific concentrations and Kd’s at physiological conditions. The authors should consider being more specific and, if possible, quantitative on this point.

Response: We have added the additional details to the text: “.. for GM-CSF, IL-3, IL-5, IL-15, IL-21, G-CSF, GH, IL-20 and IFN-ω1 (Table S4). Among them IL-3, IL-5 and GH belong to the group of cytokines with the highest affinity for S100P (Figure 4)”.

  1. I might have missed something, but, at least to me, it is not very clear to what extent the analysis of disordered regions depicted on Figure 7 adds to the conclusions of the paper. Authors should consider being more specific on the main conclusion of this figure, this possibly being that most of the cytokines analyzed seem to be quite ordered, thus validating the main conclusions of their modelling studies.

Response: Thank you for pointing this out. We extended discussion of the results of intrinsic disorder predisposition analysis and linked them to the results of modelling studies.

  1. Minor English editing is needed, especially regarding the unnecessary use of some commas (“it should be noted, that…”, “it should be emphasized, that…”, page 2, line 61; page 7, line 132; page 13, line 261) and the definite article “the” (“the both processes”, “the both mutants”, “the most of the cytokines”, among many other examples). Ca2+should be used instead of Ca2+ and ºC instead of ºC. There are also some repeated lines (page 2, line 82 to page 3, line 87) and redundancies (first paragraph on page 13).

Response: We corrected the manuscript as much as possible in accordance with these suggestions.

Reviewer 2 Report

The manuscript titled “Calcium-bound S100P Protein is a Promiscuous Binding Partner of the Four-Helical Cytokines” by Kazakov and colleagues provides data that shows the binding affinity of S100P to a variety of cytokines demonstrating that S100P can bind in the presence of Ca.  In addition, deletion of the hinge region as well as the F89A point mutant demonstrate reduced binding affinities demonstrating their importance in the interaction.  While S100P can form homodimers, the biding affinities of the cytokines indicates that biologically S100P would largely be monomeric in the blood keeping S100P to be available for cytokine binding.  A large amount of in silico work and modeling is done to reach this conclusion.

It is not clear from the manuscript what the levels of Ca are in biologically relevant situations (i.e. blood as focused on in the manuscript) in which S100P would bind to the various cytokines compared to the 1 mM CaCl2 used in the presented experiments.

A summary model figure showing S100P binding to a cytokine in the presence of Ca would help with the overall understanding of the data provided. 

Author Response

The manuscript titled “Calcium-bound S100P Protein is a Promiscuous Binding Partner of the Four-Helical Cytokines” by Kazakov and colleagues provides data that shows the binding affinity of S100P to a variety of cytokines demonstrating that S100P can bind in the presence of Ca.  In addition, deletion of the hinge region as well as the F89A point mutant demonstrate reduced binding affinities demonstrating their importance in the interaction.  While S100P can form homodimers, the biding affinities of the cytokines indicates that biologically S100P would largely be monomeric in the blood keeping S100P to be available for cytokine binding.  A large amount of in silico work and modeling is done to reach this conclusion.

It is not clear from the manuscript what the levels of Ca are in biologically relevant situations (i.e. blood as focused on in the manuscript) in which S100P would bind to the various cytokines compared to the 1 mM CaCl2 used in the presented experiments.

Response: The calcium concentration was chosen to be close to the level of free calcium in the serum of 1.1 mM (Goldstein DA. Serum Calcium. In: Walker HK, Hall WD, Hurst JW, editors. Clinical Methods: The History, Physical, and Laboratory Examinations. 3rd edition. Boston: Butterworths; 1990. Chapter 143. Available from https://www.ncbi.nlm.nih.gov/books/NBK250/). We have added this clarification to the first paragraph of section 2.1.

A summary model figure showing S100P binding to a cytokine in the presence of Ca would help with the overall understanding of the data provided.

Response: The predicted orientation of the cytokine molecule with respect to the S100P molecule is highly dependent on the cytokine, as follows from Figure 6. Therefore, the figure illustrating the summary models of S100P-cytokine complexes will look like an overlay of the numerous cytokines oriented in different directions, thereby significantly complicating perception of the data.